# Reconstructing the quantum critical fan of strongly correlated systems using quantum correlations

Irénée Frérot[1,2] & Tommaso Roscilde[2,3]

Albeit occurring at zero temperature, quantum critical phenomena have a huge impact on the finite-temperature phase diagram of strongly correlated systems, giving experimental access to their observation. Indeed, the existence of a gapless, zero-temperature quantum critical point induces the existence of an extended region in parameter space—the quantum critical fan (QCF)—characterized by power-law temperature dependences of all observables. Identifying experimentally the QCF and its crossovers to other regimes (renormalized classical, quantum disordered) remains nonetheless challenging. Focusing on paradigmatic models of quantum phase transitions, here we show that quantum correlations—captured by the quantum variance of the order parameter—exhibit the temperature scaling associated with the QCF over a parameter region much broader than that revealed by ordinary correlations. The link existing between the quantum variance and the dynamical susceptibility paves the way to an experimental reconstruction of the QCF using spectroscopic techniques.

[1] ICFO-Institut de Ciencies Fotoniques, The Barcelona Institute of Science and Technology, Av. Carl Friedrich Gauss 3, 08860 Castelldefels, (Barcelona), Spain. [2] University of Lyon, Ens de Lyon, University Claude Bernard, CNRS, Laboratoire de Physique, F-69342 Lyon, France. [3] Institut Universitaire de France, 103 Boulevard Saint-Michel, 75005 Paris, France. Correspondence and requests for materials should be addressed to I.F. (email: irenee.frerot@icfo.eu) or to T.R. (email: tommaso.roscilde@ens-lyon.fr)

Quantum critical (QC) phenomena[1–5] represent possibly the most dramatic manifestation of quantum mechanics at the macroscopic scale. Their typical setting involves an Hamiltonian $\mathcal{H} = \mathcal{H}_0 + gV$ in which the competition between the two noncommuting terms $\mathcal{H}_0$ and $V$, controlled by the parameter $g$, induces a macroscopic rearrangement of the ground state at a critical value $g_c$, accompanied by the appearance of critical quantum fluctuations of collective observables at all length scales. This complex behavior of correlation and entanglement properties emerges from extensive theoretical work based on exactly solvable microscopic models[4], quantum field theory[2], as well as numerical studies[6]. Experiments generally do not have access to ground-state physics, but it was realized at an early stage[7] that zero-$T$ quantum critical points affect a sizable portion of the finite-$T$ phase diagram, by inducing the presence of a so-called QC regime, in which the behavior of thermodynamic potentials and of response functions is controlled by the QC point (QCP). Indeed, observables in the QC regime are expected to exhibit thermal QC scaling, namely a power-law dependence on temperature with exponents descending from the critical exponents at the QCP. Strikingly, the QC regime is expected to be wider in parameter space at higher temperatures: as sketched in Fig. 1, it acts as a magnifying lens for the QCP. Even more strikingly, the finite-$T$ QC regime ignores completely the physics of the $T = 0$ and low-$T$ phases at $g \neq g_c$[8], which are generally an ordered phase with a classical analog (for, say, $g < g_c$); and a gapped quantum disordered phase (for $g > g_c$). This implies that, if the temperature is lowered from a point at $g \neq g_c$ in the QC regime, a crossover must occur toward a thermodynamic regime which is instead controlled by the presence of long-range order in the ground state—the so-called renormalized classical (RC) regime for $g < g_c$; or by the presence of a gap above a disordered ground state—the QD regime for $g > g_c$. This is all the more striking, as it shows that a strictly quantum $T = 0$ phenomenon (the QCP), governed by divergent quantum fluctuations, can have consequences on the phase diagram at temperatures $T$ which are higher than those necessary to melt long-range order via a classical thermal transition.

Many exciting platforms for the exploration of QC phenomena can be found across the physical spectrum[4,5,9–12]. But can one reconstruct the QC regime quantitatively? The special scaling properties of the thermodynamic potentials and the dynamical response functions at $g = g_c$ and finite $T$ (along the so-called QC

trajectory) have been observed in several systems, including magnetic insulators[13–17] and heavy-fermion compounds[9,18]; but the persistence of the QC regime away from $g_c$, and its crossover into the competing low-$T$ regimes, are almost uniquely observed via transport properties in heavy-fermion materials[9]—the so-called strange-metal phase in cuprate superconductors is also interpreted as an extended QC regime[19] associated with a putative QCP[20,21]. Hence it is fair to say that the quantitative extent of the QC regime, and its crossovers toward the RC and QD regimes, remain challenging to observe. Quite remarkably, the same observations can be repeated for theoretical calculations on microscopic models, for which the quantitative extent of the QC regime is rarely investigated[22]. A general scenario (corroborated by the present work) is that different observables exhibit thermal QC scaling over different regions in the $(g, T)$ parameter space. Therefore it is crucial to identify those observables which manifest such a scaling over the broadest possible range.

Here, we propose a constructive definition of the QC regime based on observables that do not admit any classical analog, namely quantum coherence measures, capturing quantum correlations and fluctuations for generic mixed states. Using quantum Monte-Carlo numerical computations and exact-diagonalization on paradigmatic models of quantum phase transitions, we single out a broad, fan-shaped region in the phase diagram where quantum fluctuations of the order parameter obey the expected power-law scaling of the QC fan. Given the link between quantum fluctuations and the dynamical susceptibility of the order parameter, our results open the way to the experimental reconstruction of the QC region using, e.g., spectroscopy on strongly correlated quantum systems.

## Results

**QC regime from quantum correlations.** At first sight, defining the QC regime using the scaling behavior of quantum fluctuations sounds very natural: the $T = 0$ QCP is characterized by critical quantum fluctuations, and the QC regime, if regarded as echoing the QCP at finite $T$, should be characterized by enhanced quantum fluctuations as well. The quantum coherence measures of our interest generally belong to the family identified by Petz[23,24] as generalizations of the quantum Fisher information (QFI)[25,26]. Among this family of quantities, we focus on the recently proposed quantum variance (QV)[27] of an observable $O$, possessing a simple definition at thermal equilibrium at inverse temperature $\beta = (k_B T)^{-1}$; it is the difference between the (total) variance (TV) $\mathrm{Var}[O] = \langle O^2 \rangle - \langle O \rangle^2$ and the susceptibility

$$\mathrm{Var}_Q[O] = \mathrm{Var}[O] - k_B T \chi_O, \tag{1}$$

where $\chi_O = (\partial \langle O \rangle / \partial h)_{h=0}$ and $h$ is a field coupling to $O$ in the Hamiltonian as $\mathcal{H} - hO$. Beyond its transparent physical meaning (difference between fluctuations and response function), the QV (like the QFI) has the fundamental property of being an entanglement witness—denying separability of the state of the system into clusters of size $k$ (or smaller) when it exceeds a $k$-dependent bound[27–30]. Moreover, unlike the QFI, it has the remarkable property of being directly accessible to state-of-the-art calculations for equilibrium quantum many-body systems at finite $T$, as, e.g., worldline quantum Monte Carlo (the computation of the QFI requires the precise knowledge of the dynamical response function at real frequencies[31], which can only be inferred from quantum Monte-Carlo data through analytical continuation, an operation very sensitive to numerical noise). This makes the QV an observable of choice to explore quantum coherence properties across the phase diagram of QC phenomena.

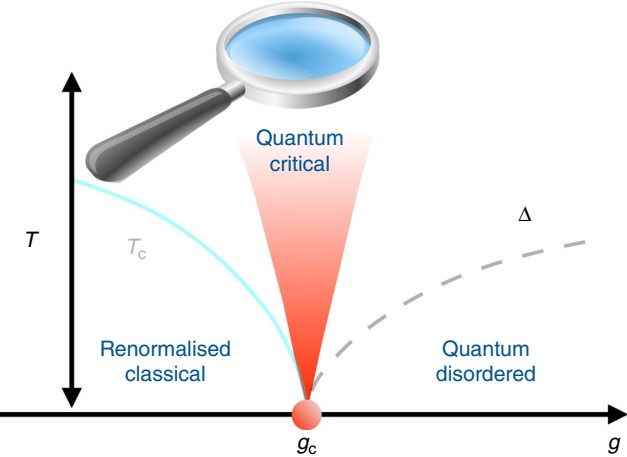

**Fig. 1** Quantum critical fan. The quantum critical fan can be seen effectively as a magnifying lens for the quantum critical point, making its existence observable over an extended range of temperatures and of the control parameter $g$ of the transition

Both the variance and the susceptibility in Eq. (1) can be expressed as integrals of the imaginary part of the dynamical susceptibility[32], resulting in the fundamental relationship:

$$\mathrm{Var}_Q[O] = \hbar \int_0^\infty \frac{d\omega}{\pi} L(\beta\hbar\omega/2)\chi_O''(\omega). \quad (2)$$

where $\mathcal{L}(x) = \coth x - 1/x$ is the Langevin function. Since $\mathcal{L}(x) \to x/3$ for $x \to 0$, one sees that $\mathrm{Var}_Q[O]$ is dominated by modes with frequency $\omega$ such that $\beta\hbar\omega \gtrsim 1$, namely modes which are mildly (or not at all) affected by thermal fluctuations. A similar expression to Eq. (2) holds for the QFI, with the replacement $\mathcal{L}(x) \to 4\tanh(x)$[31]. When $O$ is the order parameter of the quantum phase transition of interest, the dynamical susceptibility in the vicinity of the QCP is expected to obey the scaling form

$$\chi_O''(\omega) = T^{-(2-\eta)/z}\Phi_O[(g-g_c)^{\nu z}/T, \omega/T], \quad (3)$$

where $\eta$, $\nu$, and $z$ are the correlation function exponent, correlation length exponent and dynamical critical exponent of the QCP, respectively, and $\Phi_O$ is a universal function up to a prefactor[2]. This directly translates into a scaling Ansatz for the QV:

$$\mathrm{Var}_Q[O] = T^{-\psi}G_O^{(Q)}[(g-g_c)^{\nu z}/T], \quad (4)$$

where $\psi = (2-\eta)/z - 1$ and $G_O^{(Q)} \sim \int d\omega\, \mathcal{L}(\beta\hbar\omega/2)\Phi_O$. The TV of the order parameter $\mathrm{Var}[O]$ possesses a similar scaling form to Eq. (4), but with a different scaling function $G_O^{(Q)} \to G_O \sim \int d\omega\, \coth(\beta\hbar\omega/2)\Phi_O$.

Eq. (4) forms the basis of our constructive definition of the QC regime as detected by quantum correlations. Being controlled by the QCP alone, such a regime must be nearly insensitive to whether the control parameter $g$ lies above or below $g_c$. This defining condition requires that, in the QC regime, the scaling function $G_O^{(Q)}(x)$ depend very weakly on its argument, namely $G_O^{(Q)}(x) \approx G_O^{(Q)}(0)$. This leads us then to the following quantitative definition for the QC regime in the $(g, T)$ plane:

$$\mathrm{QC\ regime}: \mathrm{Var}_Q[O](g, T) \approx T^{-\psi}G_O^{(Q)}(0), \quad (5)$$

where the $\approx$ sign implies that the above condition is satisfied within some tolerance. As the QC regime is not a phase of matter which is divided from competing phases by sharp boundaries, the tolerance defines operatively the crossover lines toward the other regimes (RC and QD) in the vicinity of the QCP. The constructive definition of the QC regime offered by Eq. (5) identifies the latter regime with the region in the $g$-$T$ phase diagram in which the $T$-dependence of the quantum fluctuations of the order parameter is uniquely controlled by the presence of the QCP—namely it is the same (up to some tolerance) as along the so-called QC trajectory (the line of variable $T$ at $g = g_c$). The very fact that such a regime exists in an extended, fan-shaped region, is a fundamental test of the validity of our definition. A similar condition could obviously be formulated for the more conventional TV in the form $\mathrm{Var}[O] \approx T^{1-\frac{2-\eta}{z}}G_O(0)$: as we shall see shortly, this condition in practice singles out only the QC trajectory.

**2d transverse field Ising model**. We demonstrate our constructive definition of the QC regime using two paradigmatic examples of quantum phase transitions in quantum spin models. We begin by considering the 2d transverse field Ising (TFI)

model[4]

$$\mathcal{H}/J = -\sum_{\langle ij \rangle} S_i^z S_j^z - g\sum_i S_i^x, \quad (6)$$

where the indices $\langle ij \rangle$ and $i$ run over the nearest-neighbor bonds and the sites of a square lattice, respectively; and $S_i^\alpha$ ($\alpha = $ x, y, z) are $S = 1/2$ spin operators. A critical value $g_c = 1.522...$[33] of the transverse field divides a ferromagnetic regime ($g < g_c$) from a quantum paramagnetic one ($g > g_c$). We calculate the equilibrium properties of this model on $N = L \times L$ lattices with periodic boundary conditions numerically using Stochastic Series Expansion quantum Monte Carlo[34,35], which gives direct access to the TV and QV of most relevant observables[27]. The temperature scaling of the variances (total and quantum) of the macroscopic order parameter $J^z = \sum_i S_i^z$ in the vicinity of the QCP are contrasted in Fig. 2. Here we take for simplicity $\mathrm{Var}(J^z) = \langle (J^z)^2 \rangle$ as $\langle J^z \rangle = 0$ on finite lattices. We have checked that our conclusions do not change when considering a finite-size estimate of the actual variance, namely $\langle (J^z)^2 \rangle - N^2 m_L^2$ where $m_L^2 = \langle S_i^z S_{i+L/2}^z \rangle$. The QCP appears to control the $T$ dependence of $\mathrm{Var}(J^z)$ only along the QC trajectory [Fig. 2a], where it exhibits the expected power-law dependence $\sim T^{-\psi}$; on the contrary the TV is strongly bent upward by the finite-$T$ transition for $g < g_c$, as well as downward by the opening of a spin gap for $g > g_c$.

This picture is completely changed when one looks at quantum fluctuations. Indeed Fig. 2b shows that a power-law QC scaling of the QV as $\sim T^{-\psi}$ is manifested not only along the QC trajectory

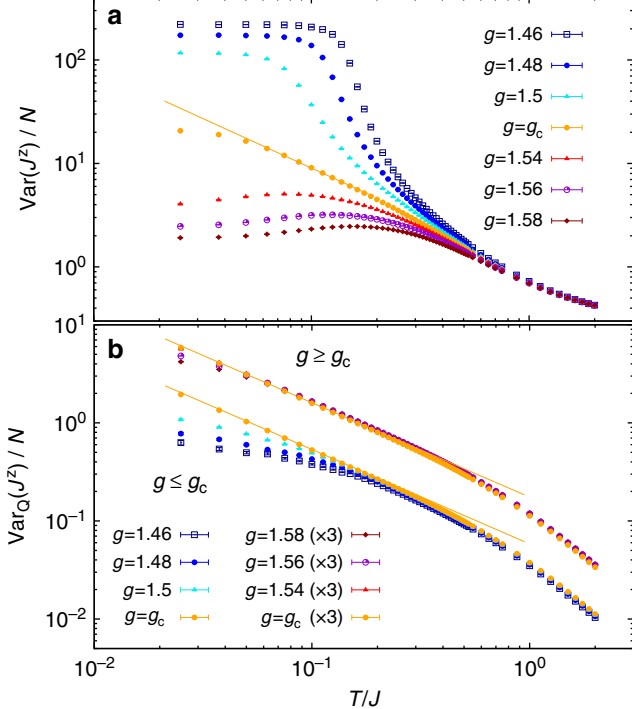

**Fig. 2** Quantum-critical scaling in the 2d transverse-field Ising (TFI) model. Temperature scaling of **a** the total variance and of **b** the quantum variance of the order parameter around the quantum critical point of the 2d TFI model (data have been obtained on a $L = 64$ lattice), for $g = 1.46$ (dark-blue squares), $g = 1.48$ (blue dots), $g = 1.5$ (light-blue triangles), $g = g_c$ (yellow dots), $g = 1.5$ (red triangles), $g = 1.2$ (purple circles), and $g = 1.54$ (dark-red diamonds). The quantum variance (**b**) above the quantum critical point ($g \geq g_c$) has been multiplied by a factor of 3 to improve readability. Solid lines are the QC scaling forms $G(0)T^{-\psi}$ (in (**a**)) and $G^{(Q)}(0)T^{-\psi}$ (in (**b**)) respectively. Here, $\psi = 0.964...$[37]

(down to $T = 0$), but it can be observed also over sizable segments of the $T$-dependence both above and below the QCP. For $g < g_c$ this is due to the fundamental property of the QV to be nearly insensitive to finite-temperature transitions[27]—as it will be discussed in a future publication, only weak singularities, in the form of inflection points, can appear at $T_c$. Therefore, the thermal critical region (in which power-law singularities dominate the behavior of the system) is minute in the $T$-dependence of the QV. Interestingly, a similar observation also applies to the case $g > g_c$ in which a finite-$T$ transition is absent. Indeed, unlike thermal fluctuations, quantum fluctuations are much more moderately suppressed by the opening of a gap. This observation can be understood by considering that, associated to the quantum fluctuations of the order parameter, there is an intrinsic quantum coherence length $\xi_Q$[36] which is always finite at finite $T$, and much smaller than the ordinary correlation length $\xi$. Approaching the QCP, $\xi \approx cT^{-1/z} = c/T$, but the opening of a gap ($\Delta$) for $g \neq g_c$ cuts off the QC growth, as $\xi$ saturates to its ground-state value $\xi(T = 0) \approx c/\Delta$. Such saturation occurs for $T \approx \Delta$. Similarly, one expects that $\xi_Q \approx c_Q/T$, but with $c_Q << c$; nonetheless, at $T = 0$, $\xi_Q$ saturates to the same value $\xi_Q(T = 0) = \xi(T = 0) = c/\Delta$. Therefore, the saturation occurs at lower temperatures, $T \approx (c_Q/c)\Delta$. In Supplementary Fig. 1, we show that $c_Q/c \approx 1/6$. Hence, one may expect that saturation temperatures for $\xi$ and $\xi_Q$ are in a similar proportion in the gapped phase. In summary, both above and below $g_c$ the QV exhibits a clear crossover from a power-law regime varying as $T^{-\psi}$ to a saturating regime, occurring around a temperature $T \sim T_c$ or $T \sim \Delta$: this behavior reveals then the crossover from the QC regime to the RC and QD regimes. We can deduce that the QCP controls the thermodynamics of quantum fluctuations over a region of sizable width, and which is fan-shaped (namely broader in $g$, the higher $T$).

In order to quantitatively reconstruct the region in which the $T$-dependence of the QV is influenced by the existence of the QCP, we establish the following criteria: (1) the QV exhibits a power-law dependence in $T$ with an exponent (namely its logarithmic derivative) reproducing $\psi = 0.964...$[37] (within some tolerance $\varepsilon$); (2) the coefficient of the power-law dependence (estimated as $\mathrm{Var}_Q(J^z)T^\psi$) reproduces the one along the QC trajectory $G^{(Q)}(0)$ (within the same tolerance $\varepsilon$). These two criteria are illustrated in Fig. 3a, b: the region matching both

criteria is then identified as the QC regime of quantum fluctuations. We would like to stress that the regions identified by criteria (1) or (2) individually can be quite different, and only their intersection can be reliably considered as representative of the QC regime. For instance criterion (2) is satisfied in a sizable portion for $g \gtrsim g_c$ at low $T$, but this is a mere coincidence due to the fact that $\mathrm{Var}_Q(J^z)T^\psi$ turns out to be a non-monotonic function, crossing twice the value $G^{(Q)}(0)$—at a lower and at a higher temperature. But only the higher-temperature crossing occurs with the QV exhibiting a logarithmic derivative compatible with $-\psi$, and hence complying with criterion 1).

Obviously the extent of such a regime in the phase diagram depends crucially on $\varepsilon$ (taken as 10% in Fig. 3): yet it is important to observe that, regardless of the value of $\varepsilon$, its lower boundaries, marking the onset of the crossovers toward the RC and QD regimes, follow faithfully the temperature scales set by $T_c$ and $\Delta$—both scaling as $|g - g_c|^{\nu z}$ in the vicinity of the QCP. In contrast, applying similar criteria to the scaling of $\mathrm{Var}(J^z)$ essentially reconstructs the QC trajectory only, as already anticipated above (Fig. 3c). In Supplementary Notes 3 and 4, we provide evidence of a similar phenomenology of the QV, as well as of the QFI, for the exactly solvable cases of the TFI in $d = 1$ (Supplementary Note 4) and $d = \infty$ (Supplementary Note 3). In the $d = 2$ case, the chosen tolerance $\varepsilon$ assigns to the QC regime a temperature extent which ranges up to a temperature $T \sim 0.3J$. This appears as a reasonable upper bound, as it remains sizeably smaller than the temperature scale $T \sim J$: at that temperature short-wavelength modes become excited, making the microscopic details of the lattice model overcome the universal power-law temperature scalings which are distinctive of the QC regime.

**Heisenberg bilayer.** We conclude by considering another paradigmatic model of quantum phase transitions, namely the $S = 1/2$ Heisenberg antiferromagnet on a square lattice bilayer, with Hamiltonian

$$\mathcal{H}/J = \sum_{\langle ij \rangle} \mathbf{S}_i \cdot \mathbf{S}_j + g \sum_{\langle lm \rangle_\perp} \mathbf{S}_l \cdot \mathbf{S}_m, \qquad (7)$$

comprising intralayer ($\langle ij \rangle$) as well as interlayer ($\langle lm \rangle_\perp$) bonds. A quantum phase transition at $g_c = 2.522...$[38] divides a Néel

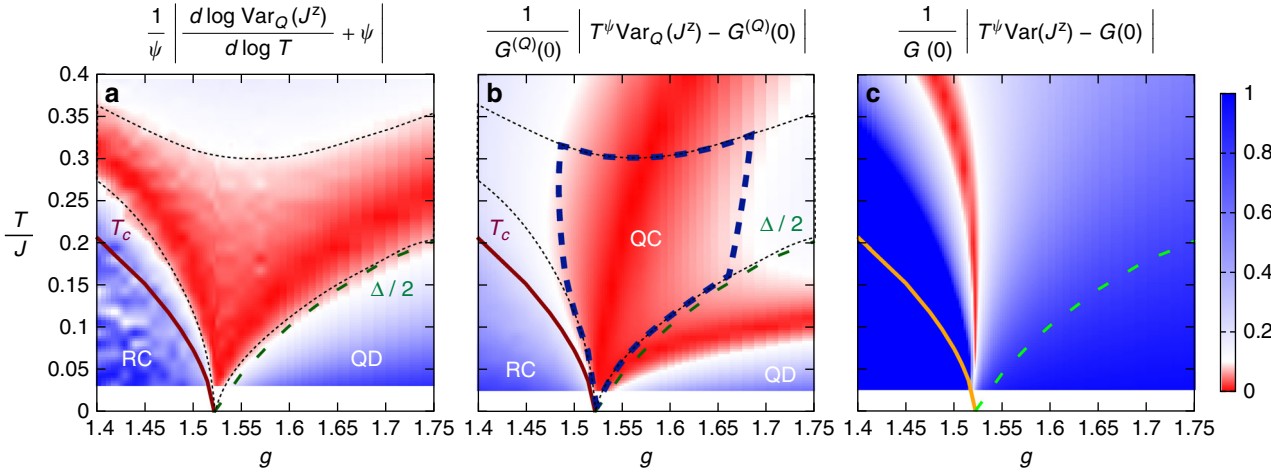

**Fig. 3** Reconstruction of the quantum critical fan of the 2d TFI model via the quantum variance. **a** Relative deviation of the logarithmic derivative of the order-parameter QV from the QC scaling exponent $\psi$—the data shown have been obtained for a $L = 64$ lattice (uneven spots in the RC region are numerical artifacts). The solid and dashed lines indicate the critical temperature[53] and half of the spectral gap (extracted from the $T$ scaling of $\langle S^x \rangle$) respectively, while the dotted line marks the region with less than 10% deviation. **b** Relative deviation of $\mathrm{Var}_Q(J^z)T^\psi$ from the QC amplitude $G^{(Q)}(0)$; same symbols as in (**a**). The dashed blue line encircles the region with less than 10% deviation on both the prefactor and on the logarithmic derivative. (**c**) Relative deviation of $\mathrm{Var}(J^z)T^\psi$ from the QC prefactor $G^{(Q)}(0)$; other symbols as in (**a**)

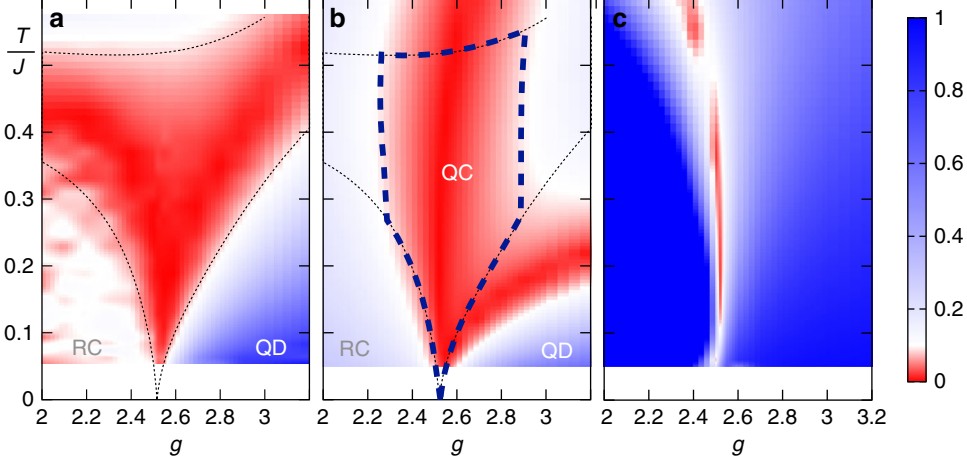

**Fig. 4** Reconstruction of the QC fan of the Heisenberg bilayer. (**a**) Relative deviation of the logarithmic derivative of the order-parameter QV from the QC scaling exponent $\psi = 0.9625$[37]. **b** Relative deviation of $\mathrm{Var}_Q(J_{st}^z)T^\psi$ from the QC amplitude $G^{(Q)}(0)$. **c** Relative deviation of $\mathrm{Var}(J_{st}^z)T^\psi$ from the QC prefactor $G^{(Q)}(0)$. Same significance of line styles as in Fig. 3, and same color scale. Uneven spots in the RC region of (**a**) are numerical artifacts. All data have been obtained on an $L \times L \times 2$ lattice with $L = 64$

antiferromagnetic regime (with order parameter given by the staggered magnetization $J_{st}^z = \sum_i (-1)^i S_i^z$) from a nonmagnetic dimer-singlet regime. The same kind of analysis of the order-parameter QV, as the one presented above for the $2d$ TFI model, leads to Fig. 4. There the QC region, where the thermal behavior of quantum fluctuations is governed by the QC point, is shown to be very broad (with a maximum width $\Delta g$ which is around 20% of $g_c$). On the contrary, a similar analysis based on the behavior of the TV singles out only a narrow region around the QC trajectory (Fig. 4c), similarly to what we reported for the TFI model. In the case at hand the QC-RC crossover is marked by a crossover in the QV from the $T^{-\psi}$ power-law behavior into another power-law behavior, as the RC regime is gapless and without a finite-$T$ phase transition; while the QC-QD crossover is similar to the one observed in the $2d$ TFI model (low-$T$ saturation of the QV). Despite its sizable width, the QC regime of the QV remains limited to a finite $g$ range, and it does not come close to the limit $g = 0$ corresponding to the most investigated case of the $2d$ Heisenberg antiferromagnet ($2d$HAF)—this holds true even when restricting uniquely to the criterion of the logarithmic derivative. The $2d$HAF Heisenberg model has been the subject of an intense search for signatures of a QC-RC crossover in the past[8,39–42]: our data (see Supplementary Fig. 2) exclude that such a crossover is visible in the QV of the order parameter.

## Discussion

Based on numerical simulations (quantum Monte Carlo) and exact diagonalization on the paradigmatic transverse-field Ising and bilayer Heisenberg models, we have provided evidence that the existence of a zero-temperature QC point (QCP) fully controls the thermodynamics of the quantum fluctuations of the order parameter (estimated via the QV) in a broad, fan-shaped region above the QCP itself. Such a region can be identified with the elusive QC regime, acting as a finite-$T$ magnifying lens of zero-$T$ quantum criticality. The extent of the QC regime, as revealed by quantum fluctuations, far exceeds that of conventional fluctuations properties—the latter contain a large thermal component subject to thermal criticality on one side of the QCP, and to a large suppression due to the opening of a gap on the other side. Therefore we open the unconventional perspective of using a property which bears no classical analog to unveil a finite-

temperature regime—somewhat reminiscent of the use of entanglement to characterize QCPs at $T = 0$.

Our proposed constructive definition of the QC regime, summarized by Eqs. (1) and (5), is completely general, and therefore it is immediately applicable to detect the QC regime in numerical as well as field-theoretical studies that have naturally access to the QV. For instance, it could be applied to unveil the QC regime of unconventional "deconfined" QC points[43–45]. Most importantly, the quantitative relationship between the dynamical susceptibility and quantum fluctuations offers the possibility to access the latter in spectroscopic experiments on strongly correlated materials. The main experimental requirement is the ability to measure the dynamical response to a probe coupling to the order parameter of the quantum phase transition; and the ability to reconstruct the corresponding dynamical susceptibility over the relevant frequency range where its imaginary part $\chi''$ has its strongest support. These requirements are clearly met by modern experiments in neutron spectroscopy[46] or AC susceptometry[47–49] on bulk materials, to cite a few relevant examples. Our proposed scheme can serve as an effective tool to unveil the existence of zero-$T$ QCPs via finite-$T$ experiments, especially in situations in which other signatures of the existence of a QCP in the low-$T$ thermodynamics prove elusive[19,50].

## Methods

**Monte Carlo calculations**. The calculations of the total and quantum variance of the order parameter shown in the main text were obtained using stochastic series expansion (SSE) quantum Monte Carlo (QMC)[34] on $N = L \times L$ lattices with periodic boundary conditions. In the case of the transverse-field Ising model the quantization axis was chosen to lie along the field ($x$) axis[35], so that the correlations among the $z$ components of the spins correspond to off-diagonal observables, which are sampled with very high statistics during the directed-loop updates[34]. A typical simulation consists of $10^4$ MC steps (each comprising as many directed loops as necessary to attempt on average to update all the spins in the SSE) for thermalization, and $10^5$ MC steps to accumulate the statistics for the observables of interest. The QV is straightforwardly obtained as the difference between the integrated equal-time correlation function and the integrated (imaginary-)time-averaged correlations[51].

## Data availability

The data supporting the findings presented in this paper, as well as the computer codes used to generate them, are available from the authors upon reasonable request.

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

## Acknowledgments

We thank L. Pezzè and A. Smerzi for fruitful discussions, and for informing us of their recent preprint on related work about the quantum Fisher information around quantum critical points[52]. This work was supported by ANR ('ArtiQ' project). Numerical simulations were run on the PSMN cluster (ENS de Lyon). I.F. acknowledges support from the Spanish Ministry MINECO (National Plan 15 Grant: FISICATEAMO No. FIS2016-79508-P, SEVERO OCHOA No. SEV-2015-0522), Fundació Cellex, Generalitat de Catalunya (AGAUR Grant no. 2017 SGR 1341 and CERCA/Program), ERC AdG OSYRIS, EU FETPRO QUIC, and the National Science Centre, Poland-Symfonia Grant no. 2016/20/W/ST4/00314.

## Author contributions

T.R. performed the quantum Monte Carlo calculations for the Ising and Heisenberg models. I.F. performed the exact-diagonalization calculations for the d = 1 and d = ∞ quantum Ising models. Both authors contributed to the writing of the manuscript.

## Additional information

**Competing interests:** The authors declare no competing interests.

