## [Peer Review File · Nature Communications]

Reviewers' comments:

Reviewer #1 (Remarks to the Author):

In this manuscript the authors propose a novel constructive prescription to quantitatively determine the extend of quantum critical regions associated with quantum phase transitions in the temperature-control parameter plane. The authors demonstrate the working of their definition by presenting numerical simulations for two model systems using Monte-Carlo simulations.

Apart from some aspects commented on in more detail below, the manuscript is well written, well structured, and well accessible also for non-experts in the field.

As the authors also emphasize, quantum critical regions (QCRs) play a key role in the understanding of the low-temperature properties of quantum many-body systems. Thus, being able to quantitatively determine QCRs is, in principle, an important and interesting aspect across a wide range of physics. In the asymptotic low-temperature limit in the vicinity of the quantum phase transition scaling theory predicts the shape of crossover lines surrounding the QCRs, but not the numerical prefactors. In the presented work the authors show that for the studied models their proposed construction reproduces the correct scaling in the low-temperature region with now providing also an estimate of the numerical prefactors. In addition, they determine crossover lines also at higher temperatures offering a possible shape of the QCR in the full temperature-control parameter plane.

While I consider the studied question, in principle, of great and fundamental interest and therefore to be suitable for publication in Nature Communications, the current version of the manuscript does not appear sufficiently convincing to me in order to support the general authors claims. In the following, let me provide the main points which led to this conclusion.

1. Although the authors show that their construction reproduces the shape of the QCR in the low-temperature limit, it does not come out clear from the presentation what are the implications of the proposed shape of the QCR at higher temperature. For example, is it the case that the proposed QCR marks those parts of the temperature-control parameter plane which are still dominated by the fixed point associated to the underlying quantum phase transition? If yes, could this, in principle, be tested?
2. The authors don't discuss in their manuscript how general their construction could be applied. For example, would it also be possible to obtain the QCR for a quantum phase transition separating two symmetry-broken phases of different kind?

In addition, there are a few further aspects which appear important to be clarified in a revised version:

1. In the abstract the authors claim that the quantum critical fan is "characterized by power-law temperature dependences of all observables". This dependence is certainly true for relevant operators, but not for any observable. For example, the transverse magnetization will presumably not. I suggest to clarify this.
2. In the introduction the authors state that in the QCR "thermodynamics is completely controlled by the quantum critical point". I would suggest to change this to thermodynamic quantities or something equivalent and one might also mention response functions such as susceptibilities.
3. In the same spirit I would suggest to change the sentence "the special scaling properties of the thermodynamics" to "the special scaling properties of the thermodynamic potentials" or equivalents.
4. In Fig. 2a the authors show numerical data for the order parameter connected correlation function. The data suggests that quantum correlations are larger in the magnetically ordered phase than at the quantum critical point, which appears unconventional to me. Is this a finite-size effect?

5. At the end of the section on "QC regime from quantum correlations" the authors claim that this "elevates the QV to the observable of choice". This appears as a rather strong statement, because it is not clear whether also other suitable quantities might exist.
6. Do the authors understand why in Fig. 3b and Fig. 4b there is such a strong quantum contribution outside the QCR on the QD side? It might be useful to comment on this in the main text in more detail.
7. From the results presented in the current manuscript some of the claims in the conclusions appear a bit too strong to me. I would suggest to mention in the first sentence that their conclusions have been drawn from numerical evidence for two model systems (and have therefore not been shown on a general level).
8. Concerning the experimental implications, the authors have included a rather short consideration, which might benefit from a more detailed discussion. Moreover, the last sentence of the conclusions suggests that it might be possible to directly observe a QCP, which, however, is anyway not possible according to the third law of thermodynamics. Thus, I would suggest to clarify this.

Summarizing, the general question addressed by the authors appears very interesting to me and worth to be considered in Nature Communications. However, a final conclusion can only be drawn provided the above critique is addressed.

Reviewer #2 (Remarks to the Author):

The authors propose a quantitative definition for the quantum critical (QC) regime of a generic quantum phase transition (driven by a parameter g at $T=0$). In a typical situation, it is known that on both sides of the quantum critical point (QCP), one finds respectively a so-called renormalized classical (RC) regime and a quantum disordered (QD) one. On both sides, there are characteristic energy scales: critical temperature (T_c) and finite gap (Δ) respectively. From this, it naturally follows that there is quantum critical fan that allows theoretically (and most importantly experimentally) to access the QCP properties at finite temperature T .

It turns out that by measuring the variance of the order parameter, the scaling form associated to the QCP is limited to a very small region in the (g, T) parameter space. Based on this observation, the authors propose to use the quantum variance (QV) of the same order parameter (a quantity that they have introduced 2 years ago, which is easily accessible numerically). Using exact solutions and numerical simulations of paradigmatic examples of QPT, they observe that QV allows to define a large critical fan regime at finite T .

I find the paper extremely well written and easy to understand. The QV quantity is related to quantum information tools but it has the advantage to possess a simple physical interpretation and to be accessible in quantum Monte-Carlo simulations. The examples that are given are nice and quite generic, so that this simple quantity could be of interest for a broad audience. For all these reasons, I recommend its publication. I only have minor comments and questions:

* I am not sure I understand why Ginzburg's criterion is not applicable, or at least why Ginzburg's region is so tiny for QV. Indeed, one may expect effective mean-field critical exponents at some distance from the QCP ?

* In Fig. 4 of the Sup. Mat., it is claimed that power-law regime is not observed at the exact QCP because of strong finite-size effects. Since it is a rather trivial model, is there a simple physical interpretation ? It should be possible to investigate size dependence in order to check this hypothesis ?

* In Fig. 5(b-c) of the Sup. Mat.: it has been shown by the authors that quantum Fisher information (QFI) satisfies: $QFI > 4 * QV$. From the plotted data, it seems to me that it is not

satisfied ? For instance $T/J=0.02$ and $g=0.48$? I know that data are obtained on different system lengths but it is still a bit misleading.

* It could be useful to also present panel (c) for Fig. 4 (such as in Fig. 3).

* When considering a finite-size estimate of the variance, one needs to compute m_L^2 . Is it given by the largest distance $(L/2, L/2)$ spin-spin correlation or by a correlation along a given axis $(L/2, 0)$ as apparently written in the text ?

* The value of the critical exponent ψ could be given in caption of Fig. 2.

* I think that it should be mandatory to provide some details about the Monte-Carlo algorithm: how many measurements ? How many thermalisation steps etc.

Reviewer #1 (Remarks to the

Author):

In this manuscript the authors propose a novel constructive prescription to quantitatively determine the extend of quantum critical regions associated with quantum phase transitions in the temperature-control parameter plane. The authors demonstrate the working of their definition by presenting numerical simulations for two model systems using Monte-Carlo simulations.

Apart from some aspects commented on in more detail below, the manuscript is well written, well structured, and well accessible also for non-experts in the field.

As the authors also emphasize, quantum critical regions (QCRs) play a key role in the understanding of the low-temperature properties of quantum many-body systems. Thus, being able to quantitatively determine QCRs is, in principle, an important and interesting aspect across a wide range of physics. In the asymptotic low-temperature limit in the vicinity of the quantum phase transition scaling theory predicts the shape of crossover lines surrounding the QCRs, but not the numerical prefactors. In the presented work the authors show that for the studied models their proposed construction reproduces the correct scaling in the low-temperature region with now providing also an estimate of the numerical prefactors. In addition, they determine crossover lines also at higher temperatures offering a possible shape of the QCR in the full temperature-control parameter plane.

While I consider the studied question, in principle, of great and fundamental interest and therefore to be suitable for publication in Nature Communications, the current version of the manuscript does not appear sufficiently convincing to me in order to support the general authors claims. In the following, let me provide the main points which led to this conclusion.

1. Although the authors show that their construction reproduces the shape of the QCR in the low-temperature limit, it does not come out clear from the presentation what are the implications of the proposed shape of the QCR at higher temperature. For example, is it the case that the proposed QCR marks those parts of the temperature-control parameter plane which are still dominated by the fixed point associated to the underlying quantum phase transition? If yes, could this, in principle, be tested?

If we understand the Reviewer correctly, (s)he is asking us whether the QCR identified via the thermal scaling of quantum variance "exhausts" (in a sense which remains to be clarified) the region of the phase diagram whose thermodynamics is governed by the existence of the critical point. In response to this remark, we would like to stress that, to the best of our understanding, there does not exist a sharp definition of the QCR - as a consequence, it is in fact very rare to find a quantitative determination of the QCR in the literature (and we make this point rather clearly in our manuscript). At a fundamental level, the absence of a sharp definition of the QCR is due to the fact that the QCR is a thermodynamic regime, and not a proper phase of matter. Its distinctive features are power-law behaviors of thermodynamics quantities with exponents related to the critical exponents of the QCP, but, as our results witness, different quantities manifest such power-law behaviors over very different ranges in the $g - T$ plane. For any given observable, the "boundaries" of the QCR are crossover regions, whose position depends on the degree of tolerance with which one requires the observable to verify the expected thermal QC scaling (compare the ϵ in Eq. (4) of the main text). Choosing different tolerance levels affects the position of these boundaries, which necessarily collapse onto the very QCP

at $T=0$ as the tolerance ϵ is sent to zero – if this were not the case, one could identify sharp boundaries of the QCR, marked by non-analiticities of finite- T observables which would be paradoxical in the absence of phase transitions. Our work shows that, for a given tolerance, the quantum fluctuations of the order parameter, quantified by its quantum variance, allow one to identify a much larger QCR than the total fluctuations of the same quantity, validating the effectiveness of our approach.

As far as the temperature extent of the QCR is concerned, the criterion based on quantum fluctuations presumably reproduces the largest temperature range over which any form of thermal QC scaling can be observed (within a given tolerance). Proving this statement rigorously would require to inspect directly the temperature dependence of all conceivable observables, something which is impossible. Yet on general grounds this is not strictly necessary, as the temperature extent that we estimate for the QCR already ranges up to $T/J \sim 0.1 - 0.5$, namely to temperature scales at which short-wavelength mode (with wavelengths comparable to the lattice spacing, and a corresponding energy $E \sim J$) can become thermally excited. As pointed out *e.g.* in Sachdev's book, beyond such energy scales one cannot hope to see the universal physics of the QCP, because the latter is only related to the long-wavelength modes which are shared with the underlying continuous-space field theory description of the lattice model of interest. Given that the Reviewer asks explicitly for a clarification of this point, we have added an extended discussion rephrasing this argument in the main text.

As a check of the above argument, in the case of the 1d transverse-field Ising model our estimate of the temperature range of the QCR matches the one offered by the temperature scaling of the free energy along the quantum-critical trajectory (this is discussed in details at the very end of the Supplementary Material). Furthermore, within the family of quantum coherence estimators that the quantum variance belongs to, tight inequalities can be established (of the kind stated in Eq. (12) of the Supplementary Material), which prevent one estimator to exhibit a very different behavior with respect to all the others. Therefore the thermal QC scaling of the quantum variance is strongly representative of the one exhibited by all other quantum coherence measures of this family (such as the quantum Fisher information, the skew information, etc.).

2. The authors don't discuss in their manuscript how general their construction could be applied. For example, would it also be possible to obtain the QCR for a quantum phase transition separating two symmetry-broken phases of different kind?

First, we wish to point out that our proposed definition of the QCR is completely general: the definition of the quantum variance given in Eq. (1) of the main text, together with the definition of the QCR given in Eq. (5), do not depend on the model. We have further stressed this point in the conclusions of the revised manuscript. The question, then, is how effective this definition is in identifying a broad, fan-shaped QCR around a quantum critical point, and this can clearly depend on the model under study. The Reviewer then suggests to apply our approach to a situation where the QCP separates two phases which break a different symmetry of the Hamiltonian. In the Ginzburg-Landau paradigm, one would generically expect such a phase transition to be of first order (excluding fine-tuning), hence featuring no QCR at all. The Reviewer may, however, have in mind the fascinating scenario of the so-called *deconfined quantum criticality* (Senthil et al., Science 2004, and Phys. Rev. B 2004) where the transition could generically be of 2nd order. The JQ-model proposed by Sandvik (Phys. Rev. Lett. 2007) is believed to realize this unconventional scenario, and exhibits quantum-critical scaling at finite temperature (Melko and Kaul, Phys. Rev. Lett. 2008). Applying our approach to unveil the QCR of this model seems indeed a very promising research direction, that we have now mentioned in our revised conclusions.

In addition, there are a few further aspects which appear important to be clarified in a revised version:

1. In the abstract the authors claim that the quantum critical fan is "characterized by power-law temperature dependences of all observables". This dependence is certainly true for relevant operators, but not for any observable. For example, the transverse magnetization will presumably not. I suggest to clarify this.

The fact that all observables acquire a power-law temperature dependence is a consequence of the vanishing of the energy gap at the QCP, which leaves the system "orphan" of any characteristic energy scale – apart from the temperature. This is per se not an exceptional condition – just consider any gapless phase (such as *e.g.* the ordered phase of the bilayer Heisenberg antiferromagnet considered in our work), which similarly exhibits power-law temperature dependence of the thermal part of any quantity at low T . The exceptional aspect of QCPs is that the power-law behavior contains the universal critical exponents of the QCP. As a matter of fact, all thermal averages can be obtained as derivatives of the free energy, and the latter possesses a part which is singular at the QCP, and which satisfies a scaling form involving the temperature in its dependence on all external fields, seen as perturbations of the QCP within the renormalization group approach. The explicit scaling form for the singular part of the free energy density, including the transverse field $g = \Gamma/J$, reads (Fisher et al., Phys. Rev. B 1989):

$$f_s(T, g) = |g - g_c|^{2-\alpha} \mathcal{G}(|g - g_c|/T^{\frac{1}{\nu z}}) = T^{\frac{d}{z}+1} \mathcal{F}(|g - g_c|/T^{\frac{1}{\nu z}}) \quad (1)$$

where we have used the quantum hyperscaling relationship $2 - \alpha = \nu(d + z)$. As $m^x = -\partial f / \partial g$, along the quantum critical trajectory $g = g_c$ one obtains

$$m^x(T) - m^x(0) = -T^{\frac{d}{z}+1-\frac{1}{\nu z}} \mathcal{F}'(0) \quad (2)$$

The above thermal scaling with an exponent $\frac{d}{z} + 1 - \frac{1}{\nu z} = 1.412\dots$ (stemming from $d = 2$, $z = 1$ and $\nu = 0.6298\dots$) is indeed exhibited in Fig. 1 by our QMC data. We take this opportunity to show the same quantity in the immediate vicinity of the quantum critical trajectory below the QCP ($g = 1.51$). The thermal QC scaling is clearly lost, as the thermal magnetisation is extremely sensitive to the appearance of a finite T_c ; the seemingly strange behaviour of the magnetisation for $g = 1.51$ is due to the fact that $m^x(T)$ is not monotonic for $g < g_c$, as it develops a maximum around T_c .

FIG. 1. Thermal quantum critical scaling of the thermal transverse magnetization in the 2d transverse field Ising model ($g = g_c$), contrasted to the same quantity for a field value below the QCP ($g = 1.51$).

2. In the introduction the authors state that in the QCR “thermodynamics is completely controlled by the quantum critical point”. I would suggest to change this to thermodynamic quantities or something equivalent and one might also mention response functions such as susceptibilities.

3. In the same spirit I would suggest to change the sentence “the special scaling properties of the thermodynamics” to “the special scaling properties of the thermodynamic potentials” or equivalents.

We thank the Reviewer for his/her suggestions; we have amended the text accordingly.

4. In Fig. 2a the authors show numerical data for the order parameter connected correlation function. The data suggests that quantum correlations are larger in the magnetically ordered phase than at the quantum critical point, which appears unconventional to me. Is this a finite-size effect?

We wish to stress that Fig 2a shows the behavior of the total variance $\langle (J^z)^2 \rangle$, containing both the quantum and the thermal contribution. The quantum contribution is plotted in Fig 2b, and is maximal along the quantum critical trajectory ($g = g_c$ with variable T), possibly matching the expectations of the Reviewer. Coming back to the total variance, instead: on the ordered side of the transition, and below T_c , it scales as N^2 on a finite-size calculation, explaining why it exceeds the total variance along the QC trajectory, whose scaling is instead extensive ($\propto N$, except at $T = 0$). If one considered a finite-size estimate of the actual variance, namely $\langle (J^z)^2 \rangle - N^2 m_L^2$ with $m_L^z = \langle S_i^z S_{i+L/2}^z \rangle$ (see also the remark to Reviewer # 2), one would find a sharp peak at T_c on the ordered side, diverging faster than the system size. As a consequence, even this estimate of the total variance in the ordered regime would exceed the same quantity along the quantum critical trajectory in some temperature range.

5. At the end of the section on “QC regime from quantum correlations” the authors claim that this “elevates the QV to the observable of choice”. This appears as a rather strong statement, because it is not clear whether also other suitable quantities might exist.

We agree with the Reviewer that a priori other observables could be suited as well. We have amended the text, stating that this “makes the QV an ideal observable”.

6. Do the authors understand why in Fig. 3b and Fig. 4b there is such a strong quantum contribution outside the QCR on the QD side? It might be useful to comment on this in the main text in more detail.

The “strong quantum contribution” that the Reviewer mentions is actually an artefact of the quantity plotted in the above cited panels. For $g > g_c$, the QV decays gradually from its ground state value upon increasing the temperature, and slower than $T^{-\psi}$ as shown e.g. in Fig. 2(b) of the main text, while it decays as T^{-2} at high T . As a consequence, $T^\psi \text{Var}_Q(J^z)$ with $0 < \psi < 1$ ($\psi = 0.964\dots$) is a quantity that increases with T at low T , and that decays as $1/T^{2-\psi}$ at high T , possessing therefore a maximum at some intermediate T . If this maximum happens to be larger than $G^{(Q)}(0)$, the quantity plotted on Fig. 3b and 4b of the main text will then vanish *twice* upon increasing T at $g > g_c$, something which is indeed realized in both models of our interest. But, in order to signal the QCR, the vanishing of the difference $T^\psi \text{Var}_Q(J^z) - G^{(Q)}(0)$ should be accompanied by a logarithmic derivative of $\text{Var}_Q(J^z)$ becoming compatible with ψ , and this is *not* the case for the region at lower T where $T^\psi \text{Var}_Q(J^z) - G^{(Q)}(0)$ nearly vanishes for $g > g_c$. Hence this region cannot be included in the QCR.

In order to clarify this aspect - which may indeed be confusing, we have added a short discussion rephrasing the above argument.

7. From the results presented in the current manuscript some of the claims in the conclusions appear a bit too strong to me. I would suggest to mention in the first sentence that their conclusions have been drawn from numerical evidence for two model systems (and have therefore not been shown on a general level).

We now specify in the beginning of our conclusion that we provide numerical evidence based on calculations made on two paradigmatic models of quantum phase transitions, and suggest (as mentioned above), that our method could be employed to reveal the QCR in a broader variety of situations.

8. Concerning the experimental implications, the authors have included a rather short consideration, which might benefit from a more detailed discussion.

We have now expanded the discussion on the experimental requirements for the measurement of the quantum variance, and added several citations to recent work on the spectroscopic study of strongly correlated bulk materials.

Moreover, the last sentence of the conclusions suggests that it might be possible to directly observe a QCP, which, however, is anyway not possible according to the third law of thermodynamics. Thus, I would suggest to clarify this.

We completely agree with the Reviewer that a direct observation of a QCP in a bulk material is prevented by the impossibility of reaching absolute zero temperature. As we state it clearly in our manuscript, the very existence of a QCR at finite temperature acts as a “magnifying lens” of the QCP at finite temperature, and makes quantum criticality observable without the need to cool the system down to absolute zero. Our last sentence states that our approach can “unveil the existence of zero- T QCPs *via finite- T experiments*”: therefore there is no contradiction with the 3rd law of thermodynamics. Yet we understand that the very concept of “direct observation of a QCP” might be disturbing, and we have rephrased it as “other signatures of the existence of a QCP in the low- T thermodynamics”.

Summarizing, the general question addressed by the authors appears very interesting to me and worth to be considered in Nature Communications. However, a final conclusion can only be drawn provided the above critique is addressed.

Reviewer #2 (Remarks to the Author):

The authors propose a quantitative definition for the quantum critical (QC) regime of a generic quantum phase transition (driven by a parameter g at $T=0$). In a typical situation, it is known that on both sides of the quantum critical point (QCP), one finds respectively a so-called renormalized classical (RC) regime and a quantum disordered (QD) one. On both sides, there are

characteristic energy scales: critical temperature (T_c) and finite gap (Δ) respectively. From this, it naturally follows that there is quantum critical fan that allows theoretically (and most importantly experimentally) to access the QCP properties at finite temperature T .

It turns out that by measuring the variance of the order parameter, the scaling form associated to the QCP is limited to a very small region in the (g, T) parameter space. Based on this observation, the authors propose to use the quantum variance (QV) of the same order parameter (a quantity that they have introduced 2 years ago, which is easily accessible numerically). Using exact solutions and numerical simulations of paradigmatic examples of QPT, they observe that QV allows to define a large critical fan regime at finite T .

I find the paper extremely well written and easy to understand. The QV quantity is related to quantum information tools but it has the advantage to possess a simple physical interpretation and to be accessible in quantum Monte-Carlo simulations. The examples that are given are nice and quite generic, so that this simple quantity could be of interest for a broad audience. For all these reasons, I recommend its publication. I only have minor comments and questions:

* I am not sure I understand why Ginzburg's criterion is not applicable, or at least why Ginzburg's region is so tiny for QV. Indeed, one may expect effective mean-field critical exponents at some distance from the QCP ?

First of all, let us clarify that for us the Ginzburg region is the region around T_c which manifests thermal criticality (namely the power-law singularities associated with thermal critical behavior), and which shrinks to zero upon moving to the QCP, as widely sketched in all finite- T phase diagrams around QCPs (see e.g. Fig 1.2 of Sachdev's book, 2nd edition). The denomination "Ginzburg region" is probably a form of jargon which is present in the literature on quantum critical points, although we understand that to a broader audience such a wording can echo the Ginzburg criterion, and that it may implicitly suggest the existence of a crossover from mean-field criticality to the actual, fluctuation-dominated criticality. We frankly have little or no numerical evidence that such a crossover is observable in the models of our interest here. To avoid any confusion, we have changed the expression "Ginzburg region" with "thermal critical region".

* In Fig. 4 of the Sup. Mat., it is claimed that power-law regime is not observed at the exact QCP because of strong finite-size effects. Since it is a rather trivial model, is there a simple physical interpretation ? It should be possible to investigate size dependence in order to check this hypothesis ?

The power-law regime is clearly observed along the QC trajectory for the quantum variance (QV) and the quantum Fisher information (QFI), but not for the total variance. The explanation is that the thermal transition line is almost vertical above the QCP. In Fig. 2(a), we show the total variance of the absolute value of the magnetization across the phase diagram for $N = 500$ spins, together with the exact value of T_c for $N = \infty$ ($T_c = g / \log[(1 + g)/(1 - g)]$).

One sees on panel (a) that the presence of the thermal phase transition precludes a clear observation of the QC scaling along $g = g_c = 1$ for the total variance. A finite-size scaling analysis, shown in Fig. 3(a), unambiguously reveals the QC region for the QV and QFI (denoted F_Q on panel (b)). The plateau (or broad shoulder) in Fig. 3(b) for the QV and the QFI marks the QC region, while the total variance exhibits no visible plateau for the sizes we considered, although it must be present in the thermodynamic limit.

FIG. 2. Ising model with infinite-range interactions for $N = 500$ spins. (a) Variance of the absolute value of the magnetization. (b) Quantum variance of the magnetization.

FIG. 3. Finite-size scaling along the QC trajectory. (a) For $T < \Delta \sim N^{-1/3}$, the variances saturate to the ground-state value $\sim N^{1/3}$. On panel (b), the temperature is rescaled to $N^{-1/3}$.

* In Fig. 5(b-c) of the Sup. Mat.: it has been shown by the authors that quantum Fisher information (QFI) satisfies: $\text{QFI} > 4^* \text{QV}$. From the plotted data, it seems to me that it is not satisfied? For instance $T/J = 0.02$ and $g = 0.48$? I know that data are obtained on different system lengths but it is still a bit misleading.

The Reviewer is right in pointing out the difference in system sizes in the two figures. Another difference is that panel (b) of Fig. 5 of the Supplementary Material was calculated for *periodic* boundary conditions, while panel (c) is for *open* boundaries. When we compute the QV for panel (c) (small size and open boundary) we do find that $\text{QFI} > 4^* \text{QV}$ (see e.g. Fig. 2(a) of our recent work, I. Frérot and T. Roscilde, Phys. Rev. Lett. 121, 020402 (2018)).

To avoid any confusion, we have added a remark in the Supplementary Material stressing out the different condition under which the two quantities (QFI and QV) have been calculated.

* It could be useful to also present panel (c) for Fig. 4 (such as in Fig. 3).

We thank the Reviewer for his/her suggestion, we have added a panel (c) to Fig. 4.

* When considering a finite-size estimate of the variance, one needs to compute m_L^2 . Is it given by the largest distance $(L/2, L/2)$ spin-spin correlation or by a correlation along a given axis $(L/2, 0)$ as apparently written in the text?

Actually the difference between the two estimates would be minor, and both estimates converge to the same value in the thermodynamic limit. The only requirement is that the spin variables sitting at distance \mathbf{r} (be it $(L/2, L/2)$ or $(L/2, 0)$) become statistically independent in the thermodynamic limit, so that $\langle S_i^z S_{i+\mathbf{r}}^z \rangle \rightarrow \langle S_i^z \rangle \langle S_{i+\mathbf{r}}^z \rangle = m^2$ – and this is guaranteed to be so as long as the separation \mathbf{r} diverges with system size. In our manuscript we opted to trade the finite-size estimate of the variance for the simple average of the square of the order parameter, as this estimate is much less affected by finite-size effects in the disordered phase (one is effectively setting $m = 0$ by hand, as expected in the thermodynamic limit). This choice allows us to exhibit the thermal QC scaling of the total variance on the system sizes we considered. Using the finite-size estimate of the variance (containing the m_L^2 term as well) would produce a curve which looks instead very different, since at low temperature (in the QCR) there is a substantial finite-size m_L for the system sizes we considered.

* The value of the critical exponent ψ could be given in caption of Fig. 2.

We have added the value of ψ in the caption, as well as in the caption of Fig. 4 of the main text.

* I think that it should be mandatory to provide some details about the Monte-Carlo algorithm: how many measurements? How many thermalisation steps etc.

We have now introduced a short Method section at the end of the main manuscript containing the main lines of the Monte Carlo calculations as well as of the exact calculations. We thank the Reviewer for prompting us to do so.

NOTE: All the modifications to the manuscript and Supplementary Material are highlighted in blue in the revised version.

REVIEWERS' COMMENTS:

Reviewer #1 (Remarks to the Author):

With their reply and with the changes made in the revised version of the manuscript, the authors have taken into account my critique almost fully except one point (Previously point 4).

This concerns the quantity plotted in Fig. 2a. The authors reply that they show the fluctuations $\langle (J_z)^2 \rangle$, from which I can immediately understand the superextensive scaling of the data in Fig. 2a in the symmetry-broken phase. The y axis label in Fig. 2a, however, is $\text{Var}(J_z) = \langle (J_z)^2 \rangle - \langle J_z \rangle^2$, as defined also above Eq. (1). For this quantity I would expect extensive scaling in system size due to general thermodynamic properties of operator fluctuations. Is it that the y axis label is just incorrect?

Provided this can be clarified I can recommend publication of the manuscript in Nature Communications.

Reviewer #2 (Remarks to the Author):

I would like to thank the authors for their rather complete and detailed answers that clarify all my previous comments.

Therefore, given the importance of proposing a new tool to access at finite temperature a quantum critical point, I recommend this paper for publication.

Reviewer #1 (Remarks to the Author):

With their reply and with the changes made in the revised version of the manuscript, the authors have taken into account my critique almost fully except one point (Previously point 4).

This concerns the quantity plotted in Fig. 2a. The authors reply that they show the fluctuations $\langle(J_z)^2\rangle$, from which I can immediately understand the superextensive scaling of the data in Fig. 2a in the symmetry-broken phase. The y axis label in Fig. 2a, however, is $\text{Var}(J_z) = \langle(J_z)^2\rangle - \langle J_z \rangle^2$, as defined also above Eq. (1). For this quantity I would expect extensive scaling in system size due to general thermodynamic properties of operator fluctuations. Is it that the y axis label is just incorrect?

Our answer: The data presented in Fig. 2 are finite-size (quantum Monte-Carlo) calculations, for which $\langle J_z \rangle = 0$, so that $\text{Var}(J_z)$ and $\langle(J_z)^2\rangle$ are rigorously equal. It is only in the thermodynamic limit that one can set $\langle J_z \rangle / N = m$ (the symmetry-broken value of the magnetization in the ordered phase). As the distribution for J_z is bi-modal in the ordered phase, with the two maxima centered around $\pm Nm$, the variance of J_z is super-extensive on any finite-size calculation.

As we explicitly write it in the text, we could have used a finite-size, extensive estimate of the variance (by subtracting $(Nm)^2$ to $\langle(J_z)^2\rangle$), which displays a peak (instead of just a sharp rise) at T_c . This finite-size estimate has the disadvantage of being numerically more noisy than $\langle(J_z)^2\rangle$, and, most importantly, of not revealing *any* form of quantum critical scaling - not even along the quantum critical trajectory! Indeed even along the QC trajectory, the quantity $(Nm)^2$ is far from zero on the finite sizes we considered. Hence using $\langle(J_z)^2\rangle$ alone allows us to get rid of annoying finite-size effects which would make the message of our figures less readable.

Provided this can be clarified I can recommend publication of the manuscript in Nature Communications.